# Development of an affirming and customizable electronic survey of sexual and reproductive health experiences for transgender and gender nonbinary people

Heidi Moseson[1]*, Mitchell R. Lunn[2,3], Anna Katz[1], Laura Fix[4], Mary Durden[1], Ari Stoeffler[5], Jen Hastings[6], Lyndon Cudlitz[7], Eli Goldberg[8], Bori Lesser-Lee[9], Laz Letcher[10], Aneidys Reyes[11], Annesa Flentje[2,12,13], Matthew R. Capriotti[2,14], Micah E. Lubensky[2], Juno Obedin-Maliver[2,15]

1 Ibis Reproductive Health, Oakland, CA, United States of America, 2 The PRIDE Study/PRIDEnet, Stanford University School of Medicine, Stanford, CA, United States of America, 3 Division of Nephrology, Department of Medicine, Stanford University School of Medicine, Stanford, California, United States of America, 4 Ibis Reproductive Health, Cambridge, Massachusetts, United States of America, 5 Planned Parenthood League of Massachusetts, Boston, Massachusetts, United States of America, 6 Department of Family and Community Medicine, University of California San Francisco, San Francisco, California, United States of America, 7 Lyndon Cudlitz Consulting, Education & Training, Albany, New York, United States of America, 8 Robert Larner M.D. College of Medicine, University of Vermont, Burlington, Vermont, United States of America, 9 Independent consultant, Malden, Massachusetts, United States of America, 10 University of New Mexico, Albuquerque, New Mexico, United States of America, 11 Edgewood College, Madison, Wisconsin, United States of America, 12 Department of Community Health Systems, University of California, San Francisco, San Francisco, California, United States of America, 13 Alliance Health Project, Department of Psychiatry, University of California, San Francisco, San Francisco, CA, United States of America, 14 Department of Psychology, San José State University, San Jose, California, United States of America, 15 Department of Obstetrics and Gynecology, Stanford University School of Medicine, Stanford, California, United States of America

* hmoseson@ibisreproductivehealth.org

**Data Availability Statement:** All relevant data are within the paper and its Supporting Information files.

## Abstract

To address pervasive measurement biases in sexual and reproductive health (SRH) research, our interdisciplinary team created an affirming, customizable electronic survey to measure experiences with contraceptive use, pregnancy, and abortion for transgender and gender nonbinary people assigned female or intersex at birth and cisgender sexual minority women. Between May 2018 and April 2019, we developed a questionnaire with 328 items across 10 domains including gender identity; language used for sexual and reproductive anatomy and events; gender affirmation process history; sexual orientation and sexual activity; contraceptive use and preferences; pregnancy history and desires; abortion history and preferences; priorities for sexual and reproductive health care; family building experiences; and sociodemographic characteristics. Recognizing that the words people use for their sexual and reproductive anatomy can vary, we programmed the survey to allow participants to input the words they use to describe their bodies, and then used those customized words to replace traditional medical terms throughout the survey. This process-oriented paper aims to describe the rationale for and collaborative development of an affirming, customizable survey of the SRH needs and experiences of sexual and gender minorities, and to

**Funding:** This study was funded by the Society of Family Planning (SFPRF11-II1) to HM; AK, AR, AS, BLL, EG, HM, JH, JOM, LC, LF, LL, and MD received partial support in the form of salaries, stipends or consulting fees from funds provided by the Society of Family Planning grant, but the Society of Family Planning did not have any additional role in the study design, data collection and analysis, decision to publish, or preparation of the manuscript. JOM was partially supported by grant K12DK111028 from the National Institute of Diabetes, Digestive, and Kidney Disorders. MRL and JOM were partially supported by Stanford University School of Medicine. MRC was supported by a Clinical Research Training Fellowship from the American Academy of Neurology and Tourette Association of America. AF was supported by National Institute on Drug Abuse K23DA039800. None of the funders had a role in the study design, nor in the collection, analysis, or interpretation of data, nor in the writing of this manuscript, nor the decision to submit the article for publication. The specific roles of funded authors are articulated in the 'author contributions' section.

**Competing interests:** AK, AR, AS, BLL, EG, HM, JH, JOM, LC, LF, LL, and MD received salary, stipend, or consulting fees from Ibis Reproductive Health via funds provided by the Society of Family Planning in a research grant (SFPRF11-II1) to HM. In addition, JOM received a one-time honorarium from Sage Therapeutics for a one-day advisory board role in May 2017, and consulted for health care provision training with Hims Inc. in 2020. This does not alter our adherence to PLOS ONE policies on sharing data and materials.

present summary demographic characteristics of 3,110 people who completed the survey. We also present data on usage of customizable words, and offer the full text of the survey, as well as code for programming the survey and cleaning the data, for others to use directly or as guidelines for how to measure SRH outcomes with greater sensitivity to gender diversity and a range of sexual orientations.

## Introduction

The ways in which we conduct research have implications for data quality and inferential value.[1] The quality and completeness of participant-reported information is intimately related to participants' direct experience.[2–4] Participant experience, in turn, is influenced by whether participants feel respected, confident in and trusting of the study investigators, and invested in the study topic.[5–10]

One way that researchers can establish trust with participants is by designing research questions that resonate with participants lived experiences. Gender identity–defined as one's internal sense of being a man, woman, both, neither of these, or something else–is a powerful determinant of one's lived experience. Gender identity can be consistent with or different from the sex that someone was assigned at birth. Sex assigned at birth is typically based on external genitalia, and is recorded as female, intersex, or male. "Transgender" is an umbrella term for people whose gender identity differs from the sex assigned to them at birth, while "cisgender" is a term for people whose gender identity aligns with their sex assigned at birth. "Nonbinary" is an umbrella term for gender identities that are not exclusively man or woman; rather, they could be a blend of both, or neither. Other words that people use for nonbinary identities include agender, bigender, gender-expansive, or genderqueer. An estimated 4.5% of the United States population, or 11.3 million people,[11] identifies as a sexual and/or gender minority (SGM).[12] At least 1.4 million transgender and gender nonbinary (TGNB) people are included in this group, and almost certainly more.[13] Gender identity and sexual orientation, however, are distinct. Gender identity refers to a person's sense of self, while sexual orientation–often labeled as being asexual, bisexual, gay, lesbian, pansexual, queer, straight or many others–encompasses how someone identifies sexually, to whom someone is attracted to romantically and or sexually, and who someone engages with sexually. Sexual orientation and its constituent domains of identity, attraction, and behavior are each independently and combined strong determinants of a person's lived experience.

Gender identity and sexual orientation are often conflated. Much sexual and reproductive health (SRH) research has made assumptions about the gender identity and sexual orientation of research participants and their sexual partners that raise concerns about data quality.[3–5, 14, 15] These problematic assumptions include: [1] research participants described as "women" explicitly include only cisgender women, thereby ignoring transgender women and nonbinary people; [2] the sexual and/or romantic partners of "women" are only cisgender men (and not cisgender women, transgender men, transgender women, nonbinary people, and/or those of another gender identity); and [3] sexual activity is assumed to refer only to sex that could lead to pregnancy or specific presentations of sexually-transmitted infections, ignoring other forms of sex that people have. Examples of these assumptions are easily found in widely used demographic, public health, and SRH surveys, both nationally and internationally. [14]

These assumptions can induce bias in SRH research in at least two ways. First, they can induce selection bias if researchers do not appropriately conceptualize the target population

and/or define eligibility criteria with sufficient detail to recruit a sample from this target population. For instance, when designing a study to evaluate risk of unintended pregnancy, the target population should include all people capable of pregnancy. However, due to lack of awareness, researchers may not consider pregnancy as a possibility for anyone other than a cisgender woman. Consequently, SRH researchers imprecisely describe eligibility criteria as "women of reproductive age" instead of more relevant criteria: the presence of a uterus in someone whose endogenous or exogenously supported hormonal milleu can carry a pregnancy. The data may systematically miss factors related to chance of pregnancy among transgender men and nonbinary people–people already known to face substantial barriers to preventative health care.[16] As a result, SRH research across subject areas may be systematically missing segments of the target population, while the health needs of a marginalized community remain inadequately characterized.

Even when SRH researchers accurately define eligibility criteria and enroll an unbiased sample, study questions that make heteronormative (*i.e.*, the belief that all people are heterosexual[17]) and cisnormative (*i.e.*, the expectation that all people are cisgender[18]) assumptions or use imprecise language about sexual activity can introduce measurement bias. As one example, the National Survey of Family Growth (NSFG) in 2015–2017 assumed involvement of a "he": *"And what about your (husband/partner) at the time? At the time you had your procedure, had **he** had all the children **he** wanted?"* [emphasis added].[19] The use of the pronoun "he" makes clear that the study investigators assume that the respondent is in a heterosexual relationship, and that the respondent's partner uses he/him/his pronouns. Modules within the national Behavioral Risk Factor Surveillance System (BRFSS) include examples of imprecision regarding sexual activity. In the 2017 Preconception Health/Family Planning module, a question asks: *"Did you or your partner do anything the last time you had sex to keep you from getting pregnant?"*[20] Given the framing of the question, the investigators were interested only in sexual activity that can lead to pregnancy. However, the question does not specify the kind of "sex." It might be interpreted in different ways depending on what "sex" means to a given participant; this could include sexual activity that leads to pregnancy and sexual activity that cannot lead to pregnancy (*e.g.*, sex between two cisgender women where no sperm is released in or near a vagina). These question design shortcomings could lead participants to [1] skip questions that seem irrelevant to their personal experiences; [2] answer a question differently than intended due to different definitions between participants and study investigators; or [3] drop out of a study that does not allow them to accurately convey their experiences or that reflects fundamental misunderstandings about their lives. Taken together, these situations could lead to more missing data, more response misclassification, or both.

As an interdisciplinary team of clinicians, researchers, and advocates, we recognize these potential biases and are concerned about their potential impact on SRH data and on participants. Because of cisnormative and heteronormative assumptions, participants may feel that SRH research is irrelevant, offensive, and erases many lived experiences, perpetuating critical knowledge gaps regarding the needs of an underserved population. Thus, we set out to co-create a survey to improve the assessment of SRH experiences of SGMs. Nearly all perinatal, contraception, and abortion research to date has focused exclusively on individuals assigned female sex at birth (AFAB) who are presumed to be cisgender and heterosexual. We sought to fill in the gaps within available research and methodologies. The objective of this process-oriented paper is to describe the collaborative development of an electronic, quantitative survey co-created by interdisciplinary research and community advisory teams to improve the relevance, precision, and affirming nature of SRH research for SGM, and to provide the full text of the final survey for others to utilize and tailor for their own research.

## Materials and methods

### Composition of study team

We formed an interdisciplinary research team of researchers with diverse gender identities and sexual orientations including a communications specialist, an epidemiologist, an obstetrician-gynecologist, a family medicine physician, an internist, qualitative researchers, a social worker, psychologists, and a reproductive health advocate. Each member of the team contributed expertise necessary for developing a customizable survey to measure and affirm SRH experiences across the gender spectrum and acknowledge a diversity of sexual orientations.

### Formative qualitative research

To inform selection and development of survey domains, we conducted 27 in-depth interviews between October 2017 and January 2018 with stakeholders in the field of SRH research and care for TGNB people AFAB. As described in detail elsewhere,[21] these stakeholders included clinicians, researchers, advocates, and patients, including those who identified as TGNB across all categories. To guide survey development, we focused analysis on responses to questions about SRH research gaps and priority SRH topics. Participants highlighted several priority issues including broader sexual health information, fertility and family building, sexually-transmitted infections, pregnancy prevention, and the need for evidence-based patient-education materials.[21]

### Recruitment and involvement of a community advisory team

In April 2018, we posted recruitment messages on social media groups (S1 File) and other community websites designed and run by TGNB people to recruit a community advisory team (CAT) for the study. The messages encouraged interested people to contact the study team. Approximately 20 candidates expressed interest; we selected five individuals to maximize CAT diversity in terms of gender identity, racial/ethnic identity, geography, and age. Included members identified as genderqueer, genderfluid, nonbinary, and transgender man, as well as Ashkenazi, Asian, Black, Latinx, and White, and resided in the Northeast, South, and Western regions of the United States. All are co-authors of this manuscript. We provided each member with information detailing the expected task and time contributions as well as the schedule for compensation. Over the 12-month survey development period, we paid each CAT member $750 for their time and expertise. CAT members participated in quarterly one-hour virtual meetings; provided high-level feedback on survey domains; provided detailed feedback on question wording, answer choices, and ordering; revised and informed recruitment strategies; and helped prioritize planned analyses.

### Iterative review and editing of survey questions

Research team expertise, findings from a literature review, formative qualitative data,[21] and consultations with CAT members informed survey domain selection. Survey domains (Table 1) and questions used and/or modified existing measures where possible from the U.S. Transgender Survey (USTS), the Behavioral Risk Factor Surveillance System (BRFSS),[22] compiled measurement work from the National Institutes of Health Sexual & Gender Minority Research Office,[23, 24] the Guttmacher 2014 Abortion Patient Survey,[25] the Nurses' Health Study 3,[26] the Growing Up Today Study,[27] Pregnancy Attitudes Timing and How (PATH) questions, Pregnancy Risk Assessment Monitoring System (PRAMS),[28] the Texas Policy Evaluation Project,[29] and guidance for clinicians regarding preconception care.[30] Within each survey domain, we created revised and/or new questions to measure the concept

**Table 1. Domains included in final quantitative survey.**

| Final quantitative survey domains |
| --- |
| 1. Current gender identity and sex assigned at birth |
| 2. Current sexual orientation |
| 3. Language used for sexual and reproductive anatomy and events |
| 4. Gender affirmation process history (hormones, surgeries) |
| 5. Sexual activity |
| 6. Contraceptive use and preferences |
| 7. Pregnancy history and desires |
| 8. Abortion history and preferences |
| 9. Priorities for sexual and reproductive health care |
| 10. Family building experiences |
| 11. Sociodemographic characteristics |

of interest without heteronormative and/or ciscentric bias in question wording (S2 and S3 Files). After finalizing the survey domains, the CAT and research team drafted the survey questions and structure. The research team then submitted survey materials to The Population Research in Identity and Disparities for Equality (PRIDE) Study (pridestudy.org) Research Advisory Committee (RAC) (pridestudy.org/team) and PRIDEnet Participant Advisory Committee (PAC) (pridestudy.org/pridenet) for review and input as part of a formal ancillary study collaboration with The PRIDE Study (pridestudy.org/collaborate). The PRIDE Study, based at Stanford University, is a community-engaged research dynamic online longitudinal cohort of SGM people that is made possible by lesbian, gay, bisexual, trans, and queer (LGBTQ+) community involvement in every step of the research process. Over approximately twelve months between May 2018 and April 2019, the study team conducted multiple rounds of revisions of survey question wording and order based on feedback from CAT members and the RAC and PAC. This work included making definitions for clinical terms more accessible, shifting the framing of questions of sexual attraction, and adding precision to questions of sexual activity (Table 2).

## Programming and testing the survey

To create a highly customized survey that could be distributed widely, we used Qualtrics (Qualtrics LLC; Provo, UT) to develop an electronic questionnaire with participant-customized language for candidate words as well as complex display and skip logic. We recognized that people use varied words for their sexual and reproductive anatomy, and that for some, the words used to describe their bodies may induce either gender dysphoria or feelings of empowerment—depending on how well the words align with a person's sense of their own bodies.[31, 32] Consequently, we programmed the survey to allow participants to input words that they use to describe their bodies, and then have those customized words replace traditional medical terms throughout the survey. In using customizable language, we aimed to create a more personalized, understandable survey that affirmed respondents' lived experiences.

To operationalize this, we programmed questions early in the survey that asked participants to provide the words they use to talk about their bodies (breasts, penis, sperm, uterus, vagina); physiological processes (menses, pregnancy); and medical procedures and treatments (abortion, contraception). The research team selected these nine customizable words because these words are known to be sensitive for particular groups, appeared frequently in the survey, and are used often by clinicians and researchers to discuss SRH issues. For each customizable word, participants indicated a preference for [1] the medical term (*i.e.*, vagina), [2] a customized word input by the participant (*i.e.*, front hole), or [3] a preference not to say (in which

**Table 2. Examples of survey question evolution as a result of iterative feedback from the Community Advisory Team (CAT).** Bolded text highlights changes in final version as compared to first draft.

| Survey Domain | First Draft | Final Draft |
|---|---|---|
| Language used for sexual and reproductive anatomy and events | *Next is a list of medical words for various body parts and experiences related to sex and fertility (the ability to get pregnant). We may ask you about these body parts in reference to your own body or to another person's body, such as a sexual partner. For each word, please let us know if you use the word listed. If you use another word, please write it in.*<br>*To improve your overall survey experience, we will use your preferred words for each of the following items whenever possible in this survey, beginning AFTER this section. We will not be able to display your own words until AFTER this section is completed.*<br>A vagina is the muscular tube that connects the external genitalia to the cervix of the uterus in most female mammals. It is the canal through which menstrual flow travels from the uterus to outside of the body.<br>• Yes, I use the word "vagina"<br>• No, I use a different word. The word I use instead of "vagina" is: ________<br>• Prefer not to say | *Next is a list of medical words for various body parts and experiences related to sex and fertility (the ability to get pregnant). We may ask you about these body parts in reference to your own body or to another person's body, such as a sexual partner. For each word, please let us know if you use the word listed. If you use another word, please write it in.*<br>*To improve your overall survey experience, we will use your preferred words for each of the following items whenever possible in this survey, beginning AFTER this section. We will not be able to display your own words until AFTER this section is completed.*<br>**A vagina is a frontal genital opening, used by some people for sexual activity, and also by some people for releasing menstrual blood or giving birth.**<br>• Yes, I use the word "vagina"<br>• No, I use a different word. The word I use instead of "vagina" is: ________<br>• Prefer not to say |
| Sexual Attraction | People are different in their sexual attraction to other people. Which best describes your feelings? Are you: *Select one.*<br>• Only attracted to women<br>• Mostly attracted to women<br>• Equally attracted to women and men<br>• Mostly attracted to men<br>• Only attracted to men<br>• Not sure | Which best describes your feelings of sexual attraction to other people? ***Select all that apply.***<br>• Attracted to women<br>• Attracted to men<br>• **Attracted to people with nonbinary identities**<br>• **Attracted to people of another gender(s) (please specify):** ___________<br>• **Not attracted to people of any gender**<br>• Not sure |
| Contraceptive use and preferences | How consistently do you use birth control?<br>• Every time I have sex<br>• Most of the times I have sex<br>• Some of the times I have sex<br>• Rarely when I have sex<br>• I never use birth control when I have sex | How consistently do you use birth control **when having sex where sperm is released in/near the** *vagina [or customized word]*?<br>• **I do not have sex where sperm is released in/near the** *vagina* **[or customized word]**<br>• Every time<br>• Most of the time<br>• Some of the time<br>• Rarely<br>• Never<br>• **I or my partner(s) have been sterilized**<br>• **I or my partner have been deemed infertile after diagnostic testing**<br>**I don't know** |

case, the medical term displayed by default) (Table 2, Row 1). We provided definitions for each customizable word that were gender-neutral and written in an accessible reading level. For those participants who provided their own word, this word was used throughout the survey each time the candidate medical term would have been used. For instance, if someone preferred "front hole" to the original candidate term "vagina," any question that used "vagina" would appear as "front hole" for that participant. Individual survey questions used up to three customizable words, which led to lengthy combinatorial display logic to ensure that each participant saw the correct words based on their stated customized words (Fig 1). We include Stata code for collapsing multiple copies of customizable-word questions to a single variable in the data cleaning phase in S3 File.

We conducted extensive survey testing to ensure that participants were displayed the correct questions based on gender identity, medical history, and customizable words. To measure current gender identity, we followed established guidelines to ask two questions: a multiple

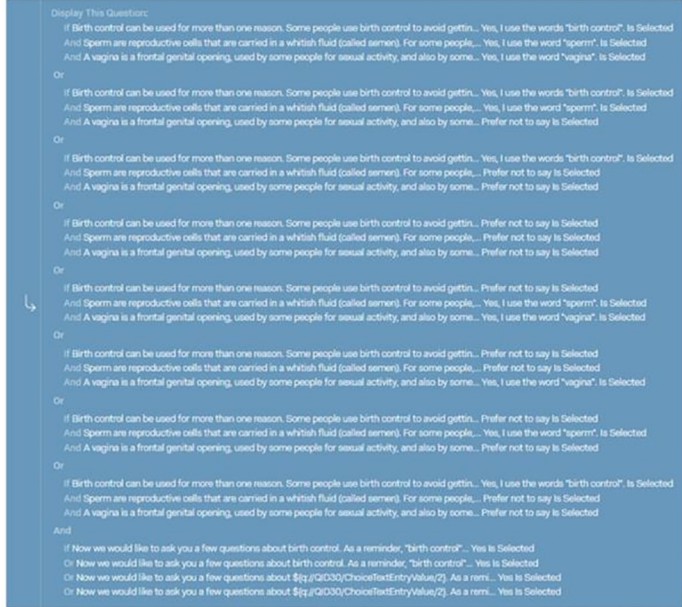

**Fig 1. Qualtrics display logic for question that includes three words for which customized piped-in word text is an option.**

choice current gender identity question and a question to assess sex assigned at birth.[33–35] However, given the difficulty of representing all gender identities in a multiple choice question, community members emphasized the importance of allowing participants to first freely self-identify with a write-in response, followed by a multiple choice "select all that apply" question, and asking about sex assigned at birth last. The final measure of gender identity that we used, first asked participants to self-identify current gender identity with an open-text question, and then to select all that apply from a list of gender identities that included: agender, cisgender man, cisgender woman, genderqueer, man, nonbinary, transgender man, transgender woman, Two-Spirit (specify if desired), woman, another gender (specify if desired), and prefer not to say. Participants then reported sex assigned at birth with answer choices: female, male, not listed (specify if desired), and prefer not to say. We went through similar processes for modifying our sexual orientation questions as we did our gender identity questions; we modified a commonly used measure of sexual orientation and expanded it to reflect a greater diversity of sexual orientations. The modified question that we used reads: "Do you consider yourself to be: asexual, bisexual, gay, lesbian, pansexual, queer, questioning, same-gender-loving, straight/heterosexual, or another sexual orientation." The survey prompted participants to select all that apply, rather than selecting a single answer from often used questions that only ask about attraction to binary gender identities.

### Recruitment of study participants

The target population for this survey included sexual and/or gender minorities (SGM) who were assigned female or intersex at birth. Eligible study participants lived in the United States or its territories, were assigned female or intersex at birth, could read and understand English, were 18 years or older and were either [1] of transgender, nonbinary, or gender-expansive experience with any sexual orientation, or [2] identified as a sexual minority cisgender woman.

We recruited participants via two approaches. First, we distributed the survey to all members of The Population Research in Identity and Disparities for Equality (PRIDE) Study (pridestudy.org). At the time of survey launch on The PRIDE Study, the cohort had 13,900 enrolled participants. The survey appeared on The PRIDE Study participant dashboard, advertised as a study on sexual and reproductive health. Any interested participant within The PRIDE Study could click on the survey and begin the screening questions for eligibility. Secondly, we also recruited participants from the general public via postings on social media, emails to community listserves, fliers at LGBTQ+ community events, and via word of mouth and boosted snowball sampling as facilitated through the social media of CAT members and their social networks. While we recruited both populations of interest through The PRIDE Study dashboard, for those recruited through the general public we limited recruitment to just TGNB individuals (not cisgender sexual minority women), and only those between the ages of 18–45 years to focus on those most likely to be of reproductive age.

### Ethical review

The Institutional Review Board at Stanford University (#: 49215, 48707) and at the University of California, San Francisco (#:18–24934) reviewed and approved the study. The PRIDE Study Research Advisory Committee (RAC) and The PRIDE Study Participant Advisory Committee (PAC) reviewed, provided input, and approved the design and conduct of this study. All participants provided written informed consent, recorded in an electronic survey form, before beginning the study survey.

## Results and discussion

### Final survey instrument

The final survey included 328 survey questions, corresponding to 1,423 variables in the dataset resulting from multiple copies of customized word questions, and multiple 'select all that apply' question structures. The final survey domains are listed in Table 1, and the survey is included in Appendix 1.

### Participant characteristics

A total of 5,005 people initiated the survey; of these, 3,110 were determined to be eligible and completed the survey (Fig 2). The majority of participants were under the age of 40 years, and reported multiple gender identities and sexual orientations (Table 3). Participants resided across the United States.

### Participant response to customized words & survey design

Across all nine customizable medical terms offered in the survey, 708 (23%) of 3,110 participants who responded to the preferred word questions provided at least one customized response, and 315 (10%) provided two or more. The three medical terms for which

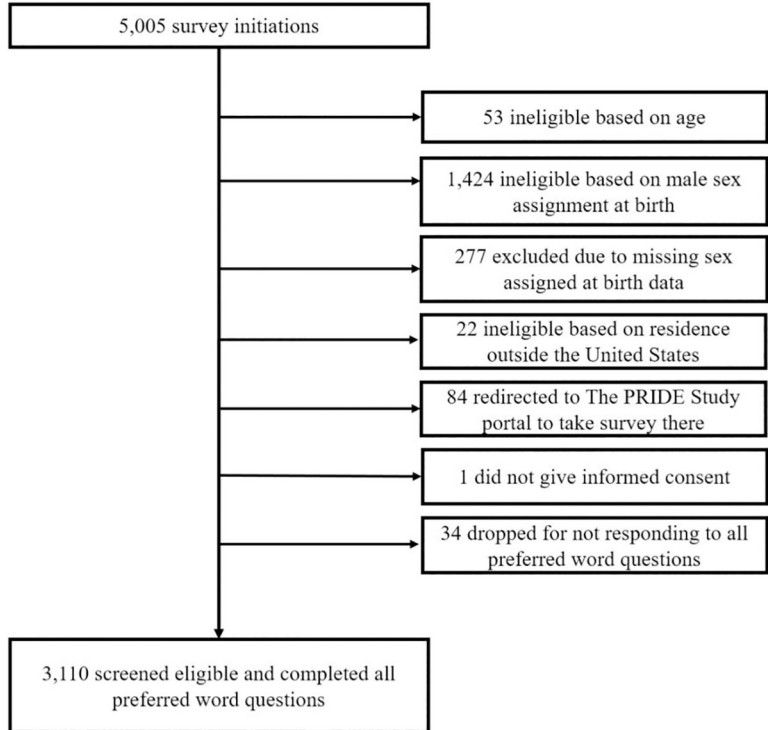

**Fig 2. Survey initiation, eligibility screening, and completion for affirming, online survey administered between April and September 2019.**

participants most frequently provided a customized word included 514 (17%) for the medical term "breasts," followed by 258 (8%) for "vagina," and 212 (7%) for "period."

In an open-ended question at the end of the survey, participants were provided space to share any feedback to the research team. Participants provided detailed feedback on study eligibility and exclusion criteria, survey content, and technical issues related to survey programming and format. Regarding eligibility criteria, some participants expressed frustration with upper age limits for the sample recruited from the general public. In terms of content, participants identified answer options that they felt were missing and expressed appreciation for question wording and the option for customizable language. Participants shared comments that highlighted the impact of the collaborative, affirming, customized nature of the survey. Selected responses are listed in Table 4.

## Conclusions

Recognizing the exclusion of SGM people from most traditional SRH research and the additional bias imposed by measurement error from imprecise survey measures, we developed a customizable, electronic, SRH-related survey to be affirming, empowering, and relevant to SGM participants. The resulting survey and lessons learned may be useful to researchers measuring health outcomes tied to sexual behavior, sexuality, and/or reproduction. In appendices, we offer the final text of the questionnaire, as well as programming details and code for cleaning the resulting data, to advance the field of survey design by creating a more inclusive and personalized research experience. Findings specific to the study research questions on the family planning needs and experiences of TGNB people, as well as cisgender sexual minority women, will be presented in manuscripts that are currently in development.

**Table 3. Sociodemographic characteristics of eligible survey participants.**

| Sample Characteristics | Eligible sample (n = 3,110) | | Sample Characteristics (continued) | Eligible sample (n = 3,110) | |
|---|---|---|---|---|---|
| Median age in years, IQR | 28 | (23, 35) | **Race/ethnicity**[*] | | |
| | | | American Indian or Alaska Native | 57 | 2 |
| | n | % | Asian, Central | 2 | 0.1 |
| **Age categories** | | | Asian, East | 75 | 2 |
| 18-19y | 276 | 9 | Asian, South | 30 | 0.1 |
| 20-24y | 750 | 24 | Asian, SouthEast | 40 | 1 |
| 25-29y | 775 | 25 | Black or African American | 108 | 3 |
| 30-34y | 530 | 17 | Hispanic or LatinX | 169 | 5 |
| 35-39y | 309 | 10 | Middle Eastern or North African | 41 | 1 |
| 40-44y | 170 | 6 | Native Hawaiian or Pacific Islander | 11 | 0.4 |
| 45-49y | 92 | 3 | White | 2716 | 87 |
| 50-54y | 71 | 2 | Unknown | 17 | 0.1 |
| 55-59y | 53 | 2 | Another race | 69 | 2 |
| 60+y | 82 | 3 | None of these | 6 | 0.1 |
| Missing | 2 | 0.1 | Missing | 149 | 5 |
| | | | **Sexual orientation**[*] | | |
| **Gender identities**[*] | | | Asexual | 374 | 12 |
| Agender | 234 | 8 | Bisexual | 1177 | 38 |
| Cisgender man[**] | 1 | 0.03 | Gay | 588 | 19 |
| Cisgender woman | 1275 | 41 | Lesbian | 872 | 28 |
| Genderqueer | 665 | 21 | Pansexual | 682 | 22 |
| Man | 293 | 9 | Queer | 1821 | 59 |
| Nonbinary | 879 | 28 | Questioning | 111 | 4 |
| Transgender man | 663 | 21 | Same gender loving | 214 | 7 |
| Transgender woman | 4 | 0.1 | Straight/heterosexual | 67 | 2 |
| Two-spirit | 26 | 1 | Another sexual orienation | 184 | 6 |
| Woman | 991 | 32 | Missing | 30 | 0.1 |
| Additional gender identity | 216 | 7 | | | |
| Prefer not to say | 2 | 0.2 | **Do you have some form of health insurance or health coverage?** | | |
| Missing | 0 | 0 | No | 145 | 5 |
| | | | Yes | 2796 | 90 |
| **Sex assigned at birth** | | | Prefer not to say | 20 | 0.6 |
| Female | 3099 | 99 | Missing | 149 | 5 |
| Not listed | 11 | 0.4 | | | |
| Missing | 0 | 0 | **US Census Region** | | |
| | | | Midwest | 573 | 18 |
| **Identifies as intersex** | | | Northeast | 685 | 22 |
| Yes | 98 | 3 | South | 646 | 21 |
| Prefer not to say | 23 | 1 | West | 846 | 27 |
| Missing | 0 | 0 | Missing | 360 | 12 |

[*] For these variables, participants could select more than one response.

[**] This participant selected "cisgender man" as their gender identity, despite selecting "female" for sex assigned at birth.

**Table 4. Selected participant responses to open-ended survey question about participant feedback on survey experience.**

| Participant feedback to the research team |
| --- |
| *"I can't tell you how much it means that you have taken such obvious and extensive efforts to be inclusive. The idea of asking participants what language they prefer to use to discuss their own bodies? Brilliant. Made me so much more comfortable taking this survey, and therefore more likely to spend a decent amount of time on it to provide thorough and thoughtful answers. I hope this becomes common practice."* |
| *"I love LOVED the use of my preferred language for body parts in the questions. After I entered that language I just assumed that it would be researched and that was that. Seeing it used to take care of me personally as a participant was really meaningful. It was a tiny way that I felt affirmed."* |
| *"I didn't realize the language questions were going to make the whole survey read so awkwardly. I would have just left it as "birth control" b/c I know what you mean by that, instead of trying to explain what language I use, which made the questions read confusingly. Also, I was kind of upset that the survey was advertised as for trans folks, but was really just for AFAB people. / / That said, I really appreciated the chance to skip over sections like the one about sexual assault. Thanks."* |
| *"I really appreciated the wide range of options available for answering most questions. It made me feel way less frustrated than most surveys where I end up checking things that don't really fit because of the lack of options."* |
| *"I recently went through an egg retrieval procedure. It was challenging on every possible level. I am disappointed that this survey did not ask any questions about fertility preservation and assisted reproduction. Aren't these a part of reproductive health too, especially for trans people?"* |
| *"Thank you for taking the time to consult with the trans community when crafting this survey, it really shows!"* |
| *"I had trouble answering some of the questions accurately. The questions about how "out" I am with different people didn't differentiate between disclosure decisions about my current gender identity vs. my trans status, which are very different."* |
| *"Thank you for this thorough, thoughtful, and affirming survey. The section asking for the language participants use to describe parts of their bodies made me misty-eyed. Please continue this important research."* |
| *"Well-phrased questions and use of my language. The most clear and affirming survey I've ever taken."* |
| *"I appreciate the care taken to avoid dysphoria triggering terms."* |

Core lessons learned included the essential role of community input from initial conceptualization to final implementation and the importance of centering the participant experience in survey design. This survey design process and resulting survey also has limitations. Engaging with multiple stakeholders and rounds of language revisions was lengthy, time-consuming, and expensive. Due to the prohibitively complex nature of programming questions with four or more customizable piped-in words, we had to restrict our questions to only three customizable terms. At some times, this artificially constrained the questions we ideally would have asked or forced the use of medical terms, even when participants had told us this was not their preference. Importantly, our survey should reduce *measurement* bias in SRH research through more inclusive and precise questions and response options. The survey in and of itself, however, does not directly address the problem of *selection* bias in SRH research. Investigators need to be mindful of gender-diversity and differences in sexual orientation when defining study eligibility criteria to directly reduce selection bias. However, the hope is that more inclusive and precise surveys will indirectly attenuate selection bias through creating more inclusive environments that foster participation from SGM participants, and simultaneously reduce drop-off from surveys once initiated. We note a number of strengths of the process and resulting questionnaire. Chiefly, the ability to use individualized, affirming, customized language for sexual and reproductive body parts and processes may avoid gender dysphoria evoked for some by medical terms. Further, modified measures of gender identity, sexual orientation, and pregnancy desires and experiences were developed to center the experiences of TGNB people (many with marginalized sexual orientations too) and to offer new, inclusive approaches to measurement of core SRH events.

Future research could expand the methodologies we utilized in a number of ways. For instance, new research could build upon the customizable word method by asking participants to use their preferred word in context by filling-in-the-blank in a sample sentence–an exercise that could lead to more specific and accurate data on words used in specific contexts. Ideally, researchers could then track substitute words with their appropriate use across settings to streamline the development of new, affirming research instruments.

The design process and final questionnaire can be used to measure epidemiological outcomes with greater sensitivity to gender diversity and diversity of sexual orientations. Future work should test the ability of these measures to reduce self-selection and non-response biases. We hope that this survey development process and resultant survey measures will inspire fellow researchers to think more inclusively and to innovate in more expansive ways to continue advancing the field of survey research, particularly for historically marginalized populations.

## Supporting information

**S1 File. Form of interest posted to recruit community advisory team members.**
(DOCX)

**S2 File. Full text of electronic customizable survey on sexual and reproductive health experiences.** Exported to Word from Qualtrics.
(PDF)

**S3 File. Qualtrics.qsf file for survey administered to people recruited through the general public.**
(QSF)

**S4 File. Sample Stata code for collapsing questions corresponding to combinations of customizable words for candidate medical terms.**
(DOCX)

## Acknowledgments

We wish to thank Brittany Charlton, Caitlin Gerdts, Sofia Filippa, Natalie Ingraham, Niara Lezama, Relebohile Motana, Sachiko Ragosta, Sarah Roberts, Samantha Ruggiero, Jane Seymour, Erin Wingo, and Yves-Yvette Young for their thoughtful contributions to this work. We also wish to thank The PRIDE Study, a community-engaged research project that serves and is made possible by LGBTQ+ community involvement at multiple points in the research process, including the dissemination of findings. We acknowledge the courage and dedication of The PRIDE Study participants, as well as all participants recruited from the general public, for sharing their stories; the careful attention of PRIDEnet Participant Advisory Committee (PAC) members for reviewing and improving every study application; and the enthusiastic engagement of PRIDEnet Ambassadors and Community Partners for bringing thoughtful perspectives as well as promoting enrollment and disseminating findings. For more information, please visit https://pridestudy.org/pridenet.

## Author Contributions

**Conceptualization:** Heidi Moseson, Anna Katz, Laura Fix, Mary Durden, Ari Stoeffler, Jen Hastings, Annesa Flentje, Micah E. Lubensky, Juno Obedin-Maliver.

**Data curation:** Heidi Moseson.

**Formal analysis:** Heidi Moseson, Anna Katz, Laura Fix.

**Funding acquisition:** Heidi Moseson, Mary Durden, Ari Stoeffler, Jen Hastings, Juno Obedin-Maliver.

**Investigation:** Heidi Moseson, Lyndon Cudlitz, Eli Goldberg, Bori Lesser-Lee, Laz Letcher, Aneidys Reyes, Micah E. Lubensky, Juno Obedin-Maliver.

**Methodology:** Heidi Moseson, Mitchell R. Lunn, Anna Katz, Laura Fix, Mary Durden, Ari Stoeffler, Jen Hastings, Lyndon Cudlitz, Eli Goldberg, Bori Lesser-Lee, Laz Letcher, Aneidys Reyes, Annesa Flentje, Matthew R. Capriotti, Juno Obedin-Maliver.

**Project administration:** Heidi Moseson, Mitchell R. Lunn, Anna Katz, Mary Durden, Annesa Flentje, Matthew R. Capriotti, Micah E. Lubensky, Juno Obedin-Maliver.

**Resources:** Heidi Moseson, Mitchell R. Lunn, Annesa Flentje, Juno Obedin-Maliver.

**Software:** Heidi Moseson.

**Supervision:** Heidi Moseson, Laura Fix, Jen Hastings, Juno Obedin-Maliver.

**Validation:** Heidi Moseson, Anna Katz, Mary Durden, Ari Stoeffler, Lyndon Cudlitz, Eli Goldberg, Bori Lesser-Lee, Laz Letcher, Aneidys Reyes.

**Writing – original draft:** Heidi Moseson, Mitchell R. Lunn.

**Writing – review & editing:** Heidi Moseson, Mitchell R. Lunn, Anna Katz, Laura Fix, Mary Durden, Ari Stoeffler, Jen Hastings, Lyndon Cudlitz, Eli Goldberg, Bori Lesser-Lee, Laz Letcher, Aneidys Reyes, Annesa Flentje, Matthew R. Capriotti, Micah E. Lubensky, Juno Obedin-Maliver.

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
