## [Decision Letter · Decision Letter 0]

4 Feb 2020

PONE-D-20-00140

Development of an affirming and customizable electronic survey of sexual and reproductive health experiences for transgender and gender non-binary people

PLOS ONE

Dear Dr. Moseson,

Thank you for submitting your manuscript to PLOS ONE. After careful consideration, we feel that it has merit but does not fully meet PLOS ONE’s publication criteria as it currently stands. Therefore, we invite you to submit a revised version of the manuscript that addresses the points raised during the review process.

We would appreciate receiving your revised manuscript by Mar 20 2020 11:59PM. To enhance the reproducibility of your results, we recommend that if applicable you deposit your laboratory protocols in protocols.io, where a protocol can be assigned its own identifier (DOI) such that it can be cited independently in the future. For instructions see: http://journals.plos.org/plosone/s/submission-guidelines#loc-laboratory-protocols

We look forward to receiving your revised manuscript.

Kind regards,

Elizabeth Ann Micks, MD, MPH

Academic Editor

PLOS ONE

Additional Editor Comments (if provided):

Specific Points for Response or Revision

Methods

--I see a description of the recruitment process for the CAT team, but I do not see a description of the

recruitment of actual survey participants. Please describe in Methods section how survey participants

were recruited both from the general public, as well as from the PRIDE study.

Results

--Line 255 – 256: The following line should be moved to the Methods section:

“We include Stata code for collapsing multiple copies of customizable 256 word questions to a single

variable in Appendix 3.”

--Though the purpose of this paper is to propose a methodology of developing personalized terms for

use in questionnaires, it would still be worthwhile to comment on the overall results of the survey

itself. It just seems odd (within the context of this paper) to say that you have surveyed 2,147 TGNB

participants, but then not make note of the actual results of that survey. I would say, at minimum, the

overall survey results should be included as a supplementary table, or if you are planning to discuss

those results in a future manuscript then make note of that in the discussion.

--Line 261 Please make a comment on the purpose of the upper age limit

Discussion

In the introduction, the authors discuss both the potential for selection bias and the potential for

measurement bias when considering research involving the gender expansive community. This study

seeks to specifically address measurement bias. However, given that selection bias is mentioned in the

introduction, it would be worthwhile to use the Discussion to briefly touch on how this applies to your

current work, or how it could be addressed in the future. The authors may even want discuss the fact

that the methodology described in this study does not address selection bias, as a potential study

weakness.

Suggestions for Future Research:

1) It may be worthwhile to ask individuals to use their words in context to address the issues with

syntax. For example, if a patient answers No and provides an alternative word to the following

prompt:

you could then follow-up with prompts such as:

Please complete the following sentence with the word you use instead of breast:

“I’ve noticed some discomfort in my left _____________”

These prompts could be tailored to potential future question in an individual study (i.e. you would use

the above prompt if you knew a future question might ask a subject about an issue with a single breast).

2) Additionally, as future iterations of this study continue a database could be created that links

particular words with their appropriate use in various grammatical settings (i.e. past, present, future

tense; possessive; plural)

3) (Line 215) it might be worthwhile to have a preference for an option that says I do not understand

this term. There is a difference between a person choosing not to select a term, and indicating that

they do not understand a term. I will note that identifying individuals who do not understand specific

terms may be of little value if the number of individuals (in the “do not understand” category) is

extremely low. For instance, for terms like vagina, most people will likely have at least a basic

understanding of what it means, rendering the “lack of understanding” choice much less useful. On

the other hand, there may be a larger number of people who do not know what the term “sperm”

means, and that information could be useful in programming you study’s decision tree.

Strengths:

--Important and relevant topic, with the potential to benefit multiple future studies involving the gender

expansive community

--Diverse study team that not only includes representation from several different subspecialties, but

individuals with diverse gender identities and sexual orientations. The former is important for identifying

the specific research needs that, in some cases, may be unique to each subspecialty. The latter is key for

all research involving the gender expansive community, especially for the development of qualitative

tools that may be used in the broader research community.

--A unique approach to addressing a challenge that is common to all researchers utilizing surveys within

the gender expansive community

Weaknesses:

--Relatively small size of the community advisory team (CAT). This is understandable (and often

unavoidable, given financial constraints associated with a compensated study) but it is worth noting that

a group of 5 individuals who may already be biased toward a more scientifically knowledgeable subpopulations (simply based on their interest and willingness to participate) could potentially introduce

bias into the study development. That said, it is commendable that you created this CAT group, and used

their input as part of this study design (an important step that many survey studies involving the gender

expansive community fail to do at all).

--Complex requirements of programming due to the multiple variations of potential subject responses

make the overall applicability of this study hard to know. For instance, simply by including terms that

would address the male sex organs in a customizable way, the programming requirements will be

increased substantially. Taking into account responses of intersex individuals will increase that

complexity even further.

Journal Requirements:

We note that one or more of the authors have an affiliation to the commercial funders of this research study : Lyndon Cudlitz Consulting Services.

Reviewers' comments:

Reviewer's Responses to Questions

**Comments to the Author**

1. Is the manuscript technically sound, and do the data support the conclusions?

Reviewer #1: Partly

Reviewer #2: Yes

2. Has the statistical analysis been performed appropriately and rigorously? 

Reviewer #1: I Don't Know

Reviewer #2: N/A

3. Have the authors made all data underlying the findings in their manuscript fully available?

Reviewer #1: Yes

Reviewer #2: Yes

4. Is the manuscript presented in an intelligible fashion and written in standard English?

Reviewer #1: Yes

Reviewer #2: Yes

6. PLOS authors have the option to publish the peer review history of their article (what does this mean?). If published, this will include your full peer review and any attached files.

Reviewer #1: Yes: Shanna K. Kattari

Reviewer #2: No

5. Review Comments to the Author

Reviewer #1: Overall, this is a very interesting topic area, and I strongly agree in the need for more measurement development and validation with and by the trans/nonbinary community. However, I found some challenges with this manuscript as detailed below.

I am a little confused as to why there are 16 authors; there should be more clarification in their roles (how many were community advocates supporting this work, statisticians, people helping with recruitment, etc). I support them being all included, and would also like to know the different ways in which they were involved. See AJPH for some examples of how this could look.

There is a need for some background info on trans/NB (TNB) individuals; who is included under this umbrella, defining cisgender, etc. *I* know what they mean, but many people reading this journal will be new to this language. The authors have the flexible word count; please use it to better ground this population.

Nonbinary is most community spelled without a hyphen; suggest switching to that.

Much of the background literature focuses on both gender and sexual orientation, including calling out of many heterocentric practices in addition to cisgenderist practices (which is great). However, the title and measurement itself seems to be focused on TNB experiences. Please remove the sexual orientation pieces to avoid the conflation of orientation with gender OR include more pieces around sexual orientation in the measure (and also be clear to name that there are LGBQ+ people who are *also* TNB to ensure the understanding of this intersection. Everything is about SGM, including the gap, until we get to the study itself, which is only about TNB people

Cite research indicating the challenges participants face (dropping out, skipping questions) when items aren't culturally responsive.

Give demographic info of the CAT participants.

I LOVE the idea of the survey allowing participants to inset words for different body parts/experiences and be used throughout. Excellent!

On page 11, I am wondering why nonbinary or enby wasn't included in the large list of terms, especially given the percentage this group has represented in USTS and other studies, as well as this manuscript using this language of nonbinary so frequently.

I may be missing it, but I don't see the demographics of the participants. What was the break down around gender, race, age, disability status, etc.?

I was actually really shocked when I got to the end and there wasn't a measure. Most of the writing made this feel like this manuscript was about the developing of a measure, so I excepted to see one. As is, it feels incredibly incomplete, like it was written for a class project or grant without the study being done. I would highly recommend either 1) re-writing much of the manuscript to be clear that this is a process paper, specifically looking at the experiences of participants filling in their own language, OR 2) Given that this journal has a more flexible word limit, consider including the analysis and development of the actual measure.

As is, this feels pretty confusing and incomplete, but the topic is so important and I would love to see this as a strong article to make solid impact. I look forward to this measure being made available for us!

---

## [Author Response · Author response to Decision Letter 0]

26 Mar 2020

PONE-D-20-00140

Development of an affirming and customizable electronic survey of sexual and reproductive health experiences for transgender and gender non-binary people

PLOS ONE

We are grateful to the reviewers and editorial team for these thoughtful comments. We have responded point by point below, with line numbers corresponding to the clean (non-track changes) version of the revised manuscript.

Response to Reviewers

Additional Editor Comments (if provided):

Specific Points for Response or Revision

Methods

--I see a description of the recruitment process for the CAT team, but I do not see a description of the recruitment of actual survey participants. Please describe in Methods section how survey participants were recruited both from the general public, as well as from the PRIDE study.We have added a section to the Methods called “Recruitment of study participants” where we have moved the information about study eligibility (which was previously in the results section), and also added a description of how we recruited participants. The added text in lines 289-307 reads: 

“The target population for this survey included sexual and/or gender minorities (SGM)(12) of who were assigned female or intersex at birth. Eligible study participants lived in the United States or its territories, were assigned female or intersex at birth, could read and understand English, were 18 years or older. and were either [1] of transgender, nonbinary, or gender-expansive experience with any sexual orientation, or [2] identified as a sexual minority cisgender woman. 

We recruited participants via two approaches. First, we distributed the survey to all members of The Population Research in Identity and Disparities for Equality (PRIDE) Study (pridestudy.org). At the time of survey launch on The PRIDE Study, the cohort had 13,900 enrolled participants. The survey appeared on The PRIDE Study participant dashboard, advertised as a study on sexual and reproductive health. Any interested participant within The PRIDE Study could click on the survey and begin the screening questions for eligibility. Secondly, we also recruited participants from the general public via postings on social media, emails to community listserves, fliers at LGBTQ+ community events, and via word of mouth and boosted snowball sampling as facilitated through the social media of CAT members and their social networks. While we recruited both populations of interest through The PRIDE Study dashboard, for those recruited through the general public we limited recruitment to just TGNB individuals (not cisgender sexual minority women), and only those between the ages of 18-45 years to focus on those most likely to be of reproductive age.”

Results

--Line 255 – 256: The following line should be moved to the Methods section:

“We include Stata code for collapsing multiple copies of customizable 256 word questions to a single variable in Appendix 3.”

We have moved this line to the Methods section, specifically in the section on programming and testing the survey (line 258-260).

--Though the purpose of this paper is to propose a methodology of developing personalized terms for use in questionnaires, it would still be worthwhile to comment on the overall results of the survey itself. It just seems odd (within the context of this paper) to say that you have surveyed 2,147 TGNB participants, but then not make note of the actual results of that survey. I would say, at minimum, the overall survey results should be included as a supplementary table, or if you are planning to discuss those results in a future manuscript then make note of that in the discussion.

This is a point well-taken. The paper is intended to be about the rationale for and process of developing the survey, and to share the survey itself, rather than reporting on results of the survey. Future manuscripts are focused on the many different results of the study. However, given that we do have the study results, we have added Figure 2 with a flowchart of participant survey initiation and eligibility screening, as well as Table 3 with findings on participant sociodemographic characteristics, as well as included some high-level findings related to the customized word responses in the relevant section in the results. In addition, we have added a sentence to the end of the first paragraph of the discussion to make clear that other results are forthcoming. The added text in line 360-362 reads: “Findings specific to the study research questions on the family planning needs and experiences of transgender and gender nonbinary people, as well as cisgender sexual minority women, will be presented in manuscripts that are currently in development.”

--Line 261 Please make a comment on the purpose of the upper age limit

We have added a great deal of text in the methods section regarding participant recruitment (as quoted in an earlier response above). We included the following line (304-307) in an attempt to clarify the upper age limit for people recruited through the general public: “While we recruited both populations of interest through The PRIDE Study dashboard, for those recruited through the general public, we limited recruitment to just TGNB individuals (not cisgender sexual minority women), and only those between the ages of 18-45 years to focus on those most likely to be of reproductive age.”

Discussion

In the introduction, the authors discuss both the potential for selection bias and the potential for

measurement bias when considering research involving the gender expansive community. This study seeks to specifically address measurement bias. However, given that selection bias is mentioned in the introduction, it would be worthwhile to use the Discussion to briefly touch on how this applies to your current work, or how it could be addressed in the future. The authors may even want discuss the fact that the methodology described in this study does not address selection bias, as a potential study weakness.

This is an excellent point. We have added text in the discussion section to address this. The added text in lines 371-378 reads: “Importantly, our survey should reduce measurement bias in SRH research through more inclusive and precise questions and response options. The survey in and of itself, however, does not directly address the problem of selection bias in SRH research. Investigators need to be mindful of gender-diversity and differences in sexual orientation when defining study eligibility criteria to directly reduce selection bias. However, the hope is that more inclusive and precise surveys will indirectly attenuate selection bias through creating more inclusive environments that foster participation from SGM participants, and simultaneously reduce drop-off from surveys once initiated.”

Suggestions for Future Research:

1) It may be worthwhile to ask individuals to use their words in context to address the issues with

syntax. For example, if a patient answers No and provides an alternative word to the following

prompt: you could then follow-up with prompts such as: Please complete the following sentence with the word you use instead of breast: “I’ve noticed some discomfort in my left _____________” These prompts could be tailored to potential future question in an individual study (i.e. you would use the above prompt if you knew a future question might ask a subject about an issue with a single breast).

This is an excellent suggestion, and one that we wish we had incorporated because it may have helped participants to be more succinct in the words they provided. We have added a paragraph on suggestions for future research to the discussion, and specifically suggest the above idea. The new text in lines 385-388 reads: “Future research could expand the methodologies we utilized in a number of ways. For instance, new research could build upon the customizable word method by asking participants to use their preferred word in context by filling-in-the-blank in a sample sentence – an exercise that could lead to more specific and accurate data on words used in specific contexts.”

2) Additionally, as future iterations of this study continue a database could be created that links

particular words with their appropriate use in various grammatical settings (i.e. past, present, future tense; possessive; plural)

This is also an interesting idea, and one we will keep in mind as we move forward with this work. We have added text to the above sentence to highlight this idea. The new text in lines 388-390 reads: “Ideally, researchers could then track substitute words with their appropriate use across settings to streamline the development of new, affirming research instruments.”

3) (Line 215) it might be worthwhile to have a preference for an option that says I do not understand this term. There is a difference between a person choosing not to select a term, and indicating that they do not understand a term. I will note that identifying individuals who do not understand specific terms may be of little value if the number of individuals (in the “do not understand” category) is extremely low. For instance, for terms like vagina, most people will likely have at least a basic understanding of what it means, rendering the “lack of understanding” choice much less useful. On the other hand, there may be a larger number of people who do not know what the term “sperm” means, and that information could be useful in programming you study’s decision tree.

This is a great comment, and a challenge that we came up against in the pilot testing of the survey. As a result, realizing that not everyone knows the meaning of the words “sperm” or “uterus”, for example, we worked with our community advisory team to develop gender-neutral definitions for all of the words that were asked. This helped ensure that people would have at least a basic understanding of the word’s meaning.

Strengths:

--Important and relevant topic, with the potential to benefit multiple future studies involving the gender expansive community

--Diverse study team that not only includes representation from several different subspecialties, but individuals with diverse gender identities and sexual orientations. The former is important for identifying the specific research needs that, in some cases, may be unique to each subspecialty. The latter is key for all research involving the gender expansive community, especially for the development of qualitative tools that may be used in the broader research community.

--A unique approach to addressing a challenge that is common to all researchers utilizing surveys within the gender expansive community

Thank you!

Weaknesses:

--Relatively small size of the community advisory team (CAT). This is understandable (and often

unavoidable, given financial constraints associated with a compensated study) but it is worth noting that a group of 5 individuals who may already be biased toward a more scientifically knowledgeable subpopulations (simply based on their interest and willingness to participate) could potentially introduce bias into the study development. That said, it is commendable that you created this CAT group, and used their input as part of this study design (an important step that many survey studies involving the gender expansive community fail to do at all).

Thank you for pointing this out. Given the tremendous contributions of the CAT to this study, we feel similarly about the value of involving even more people in future work. Moving forward, we will aim to budget for more CAT members so that we minimize the potential bias outlined above. We also failed to highlight in our initial draft the tremendous contributions of The PRIDE Study Research Advisory Council and Participant Advisory Council that also reviewed all study materials and provided essential input. These groups include an even larger array of diversity in terms of gender identity and sexual orientation. The added text in lines 212-223 reads: “After finalizing the survey domains, the CAT and research team drafted the survey questions and structure. The research team then submitted survey materials to The Population Research in Identity and Disparities for Equality (PRIDE) Study (pridestudy.org) Research Advisory Committee (RAC) (pridestudy.org/team) and PRIDEnet Participant Advisory Committee (PAC) (pridestudy.org/pridenet) for review and input as part of a formal ancillary study collaboration with The PRIDE Study (pridestudy.org/collaborate). The PRIDE Study, based at Stanford University, is a community-engaged research dynamic online longitudinal cohort of SGM people that is made possible by lesbian, gay, bisexual, trans, and queer (LGBTQ+) community involvement in every step of the research process. Over approximately twelve months between May 2018 and April 2019, the study team conducted multiple rounds of revisions of survey question wording and order based on feedback from CAT members and the RAC and PAC.”

--Complex requirements of programming due to the multiple variations of potential subject responses make the overall applicability of this study hard to know. For instance, simply by including terms that would address the male sex organs in a customizable way, the programming requirements will be increased substantially. Taking into account responses of intersex individuals will increase that complexity even further.

Agreed. We recognize the complexity of the survey programming, and in an attempt to make it more accessible for peers, offer the full text of the survey, including the .qsf file for the Qualtrics survey, in an appendix so that others can more easily modify/replicate the survey design.

Journal Requirements:

Based on our review, our manuscript is in compliance with PLOS ONE’s style requirements.

We note that one or more of the authors have an affiliation to the commercial funders of this research study: Lyndon Cudlitz Consulting Services.

Lyndon Cudlitz, an author on this paper, confirmed that the Society of Family Planning was listed as a funder of their work – but LC clarified in an email this week that this reference to SFP was only meant in reference to Ibis Reproductive Health having received a grant from the Society of Family Planning, from which LC and other co-authors were paid a stipend or consulting fee. We have amended both the competing interests and funding statements to reflect this.

We have revised the Funding Statement to reflect this, and will update the author roles in the Author Contributions section as well.

We have added this statement to the Funding Statement.

The commercial affiliation did NOT play a role in our study, and this is reflected in our Funding Statement (included in cover letter as well).

We have updated the Competing Interests statement to now read: “All authors received salary, stipend, or consulting fees from Ibis Reproductive Health via funds provided by the Society of Family Planning in a research grant (SFPRF11-II1) to HM. In addition, JOM received a one-time honorarium from Sage Therapeutics for a one-day advisory board role in May 2017, and consulted for health care provision training with Hims Inc. in 2020. This does not alter our adherence to PLOS ONE policies on sharing data and materials. Rather, our adherence to PLOS ONE policies on sharing data and materials are informed by other experiences. Members of the lesbian, gay, bisexual, transgender, and queer (LGBTQ) communities have experienced significant stigma and discrimination from society including the medical and investigational communities. As such, we are ethically bound to upholding the principle of non-maleficence; we promise our participants to not let any data (including de-identified) fall into the hands of people who may use it to publish stigmatizing results about the LGBTQ communities. For example, someone could look at the gender identities of racial/ethnic minorities and make claims that a specific racial/ethnic minority should be targeted to ‘cure’ people with a specific gender identity. As such, we have a developed an Ancillary Study process in which investigators interested in using our data submit a brief application which is reviewed by both a Research Advisory Committee (composed of scientists) and Participant Advisory Committee (composed of participants) to affirm appropriate data use. Details about the Ancillary Study process are available at www.pridestudy.org/collaborate or by contacting us at support@pridestudy.org or 855-421-9991 (toll-free).”

We have included these statements in our cover letter. Thank you for changing the online submission on our behalf. 

Reviewer #1: Overall, this is a very interesting topic area, and I strongly agree in the need for more measurement development and validation with and by the trans/nonbinary community. However, I found some challenges with this manuscript as detailed below.

I am a little confused as to why there are 16 authors; there should be more clarification in their roles (how many were community advocates supporting this work, statisticians, people helping with recruitment, etc). I support them being all included, and would also like to know the different ways in which they were involved. See AJPH for some examples of how this could look.

We thank the reviewer for this comment. The sixteen authors include the study co-investigators, the project managers and research assistants that worked on the study, five community advisory team members, three research advisory committee (RAC) members from The PRIDE Study, and the co-director of The PRIDE Study who worked closely with us on the survey. Each of the 16 authors contributed to authorship in at least three categories, including conceptualization of the study, grant-writing to obtain funding for the work, design of the survey itself, and drafting and reviewing of the final manuscript. We have described the contributions of each in the online PLOS ONE “Authorship contributions” section, drawing from the language/style of AJPH (thank you for the suggestion).

There is a need for some background info on trans/NB (TNB) individuals; who is included under this umbrella, defining cisgender, etc. *I* know what they mean, but many people reading this journal will be new to this language. The authors have the flexible word count; please use it to better ground this population.

This is an excellent point. We have added definitions to the main body of the text to provide more clarity, as this may be new to some readers. The new definitions are in lines 85-104, and include some text that was previously only in a footnote, as well as added text. We specifically added definitions for sexual orientation as well, and an explicit note that the two are distinct (per a later comment from this reviewer). The added text now reads: “One way that researchers can establish trust with participants is by designing research questions that resonate with participants lived experiences. Gender identity – defined as one’s internal sense of being a man, woman, both, neither of these, or something else – is a powerful determinant of one’s lived experience. Gender identity can be consistent with or different from the sex that someone was assigned at birth. Sex assigned at birth is typically based on external genitalia, and is recorded as female, intersex, or male. “Transgender” is an umbrella term for people whose gender identity differs from the sex assigned to them at birth, while “cisgender” is a term for people whose gender identity aligns with their sex assigned at birth. “Nonbinary” is an umbrella term for gender identities that are not exclusively man or woman; rather, they could be a blend of both, or neither. Other words that people use for nonbinary identities include agender, bigender, gender-expansive, or genderqueer. An estimated 4.5% of the United States population, or 11.3 million people,(11) identifies as a sexual and/or gender minority (SGM)(12). At least 1.4 million transgender and gender nonbinary (TGNB) people are included in this group, and almost certainly more.(13) Gender identity and sexual orientation, however, are distinct. Gender identity refers to a person’s sense of self, while sexual orientation – often labeled as being asexual, bisexual, gay, lesbian, pansexual, queer, straight or many others – encompasses how someone identifies sexually, to whom someone is attracted to romantically and or sexually, and who someone engages with sexually. Sexual orientation and its constituent domains of identity, attraction, and behavior are each independently and combined strong determinants of a person’s lived experience.”

Nonbinary is most community spelled without a hyphen; suggest switching to that.

Thank you for this guidance, we are happy to change to “nonbinary” without the hyphen. 

Much of the background literature focuses on both gender and sexual orientation, including calling out of many heterocentric practices in addition to cisgenderist practices (which is great). However, the title and measurement itself seems to be focused on TNB experiences. Please remove the sexual orientation pieces to avoid the conflation of orientation with gender OR include more pieces around sexual orientation in the measure (and also be clear to name that there are LGBQ+ people who are *also* TNB to ensure the understanding of this intersection. Everything is about SGM, including the gap, until we get to the study itself, which is only about TNB people.

The reviewer makes a thoughtful point. We are well aware of the all too common conflation of gender identity and sexual orientation. The survey was geared toward TGNB people assigned female at birth with any sexual orientation, and also cisgender sexual minority women – so we did a lot in the survey to address gender identity AND sexual orientation. However, we appreciate that this did not come through clearly in the first submission. We have added language to make this clearer from the outset, and have included an example where relevant of measures focused on sexual orientation. The added example on sexual orientation/identity is included in the methods in lines 278-285, and reads: “We went through similar processes for modifying our sexual orientation questions as we did our gender identity questions; we modified a commonly used measure of sexual orientation and expanded it to reflect a greater diversity of sexual orientations. The modified question that we used reads: “Do you consider yourself to be: asexual, bisexual, gay, lesbian, pansexual, queer, questioning, same-gender-loving, straight/heterosexual, or another sexual orientation.” The survey prompted participants to select all that apply, rather than selecting a single answer from often used questions that only ask about attraction to binary identities.”

Cite research indicating the challenges participants face (dropping out, skipping questions) when items aren't culturally responsive.

Thank you for this suggestion. We have added references to the first paragraph of the introduction, where we make claims about the importance of the research process reflecting participants’ experiences, and also treating participants with respect. The references we cite draw from the authors’ own experiences implementing novel survey methodologies to improve data quality, as well as a broader literature on the failures and successes in the development and assessment of measures to capture elements of SGM identity and health.

Give demographic info of the CAT participants.

We have added information on gender and racial identities, as well as region of residence, to the section on the CAT. The new text now reads: “Included members identified as genderqueer, genderfluid, nonbinary, and trans man, as well as Ashkenazi, Asian, Black, Latinx, and White, and resided in the East, North, South, and Western regions of the United States. All are co-authors of this manuscript.”

I LOVE the idea of the survey allowing participants to inset words for different body parts/experiences and be used throughout. Excellent!

Thank you. We did too! And so did participants. 

On page 11, I am wondering why nonbinary or enby wasn't included in the large list of terms, especially given the percentage this group has represented in USTS and other studies, as well as this manuscript using this language of nonbinary so frequently.

Thank you for catching this. We did include “Nonbinary” as an option in the actual survey – but failed to list it in the narrative text of the manuscript – so thank you for flagging! We have added it now. The revised text now reads: “The final measure of gender identity that we used, first asked participants to self-identify current gender identity with an open-text question, and then to select all that apply from a list of gender identities that included: agender, cisgender man, cisgender woman, genderqueer, man, nonbinary, transgender man, transgender woman, Two-Spirit (specify if desired), woman, another gender (specify if desired), and prefer not to say.”

I may be missing it, but I don't see the demographics of the participants. What was the break down around gender, race, age, disability status, etc.?

As this was intended to be a study design paper, we did not initially include demographic data on participants. Most of this information will be included in forthcoming publications. The objective of this paper is to describe the survey design process, and provide the text of the survey itself. However, given feedback from the reviewers, we have reconsidered and have included some brief results on participants with age, gender identities, racial identities, and region included. This information is now included in Table 3, as well as narratively described in the results. We also added a Figure 2 to show in a flow chart the number of people that initiated the survey, and why/how people were excluded during the screening process. The added text in lines 323-324 reads: “A total of 5,005 people initiated the survey; of these, 3,110 were determined to be eligible and completed the survey (Figure 2). The majority of participants were under the age of 40 years, and reported multiple gender identities and sexual orientations (Table 3). Participants resided across the United States.” 

I was actually really shocked when I got to the end and there wasn't a measure. Most of the writing made this feel like this manuscript was about the developing of a measure, so I excepted to see one. As is, it feels incredibly incomplete, like it was written for a class project or grant without the study being done. I would highly recommend either 1) re-writing much of the manuscript to be clear that this is a process paper, specifically looking at the experiences of participants filling in their own language, OR 2) Given that this journal has a more flexible word limit, consider including the analysis and development of the actual measure. As is, this feels pretty confusing and incomplete, but the topic is so important and I would love to see this as a strong article to make solid impact. I look forward to this measure being made available for us!

Thank you for this important feedback. This paper is intended to be a process paper – about the rationale for and process of designing a more inclusive and affirming survey. We do, however, also present and share the measures we developed/modified in the full survey instrument, which we include and will publish as an appendix to the paper. That way, other researchers can view all of the survey questions directly and use/modify to suit their research question. This is motivated by a desire to make it as easy as possible for other researchers to learn from and build off of the process we followed, given that not all teams might have the time to invest in the process we followed. We attempted to make this objective clear with the title of the paper, the abstract, and in the last paragraph of the introduction (lines 157-165), with the following text: “Thus, we set out to co-create a survey to improve the assessment of SRH experiences of SGMs. Nearly all perinatal, contraception, and abortion research to date has focused exclusively on individuals assigned female sex at birth (AFAB) who are presumed to be cisgender and heterosexual. We sought to fill in the gaps within available research and methodologies. The objective of this process-oriented paper is to describe the collaborative development of an electronic, quantitative survey co-created by interdisciplinary research and community advisory teams to improve the relevance, precision, and affirming nature of SRH research for SGM, and to provide the full text of the final survey for others to utilize and tailor for their own research.” The modified text in the abstract in lines 69-75 now reads: “This process-oriented paper aims to describe the rationale for and collaborative development of an affirming, customizable survey of the SRH needs and experiences of sexual and gender minorities, and to present summary demographic characteristics of 3,110 people who completed the survey. We also present data on usage of customizable words, and offer the full text of the survey, as well as code for programming the survey and cleaning the data, for others to use directly or as guidelines for how to measure SRH outcomes with greater sensitivity to gender diversity and a range of sexual orientations.” We have also made edits throughout the manuscript to try to make this clearer, so that other readers are not similarly misled. We welcome any suggestions for doing this more clearly than what we’ve done.

Thank you for this helpful tool! We have run both figures through PACE and downloaded the approved/green check-marked versions and will upload the PACE-approved/formatted images with the manuscript.

---

## [Editor Report · Decision Letter 1]

9 Apr 2020

Development of an affirming and customizable electronic survey of sexual and reproductive health experiences for transgender and gender nonbinary people

PONE-D-20-00140R1

Dear Dr. Moseson,

We are pleased to inform you that your manuscript has been judged scientifically suitable for publication and will be formally accepted for publication once it complies with all outstanding technical requirements.

With kind regards,

Elizabeth Ann Micks, MD, MPH

Academic Editor

PLOS ONE

---

## [Editor Report · Acceptance letter]

20 Apr 2020

PONE-D-20-00140R1 

Development of an affirming and customizable electronic survey of sexual and reproductive health experiences for transgender and gender nonbinary people 

Dear Dr. Moseson:

I am pleased to inform you that your manuscript has been deemed suitable for publication in PLOS ONE. Congratulations! Your manuscript is now with our production department. 

With kind regards,

on behalf of

Dr. Elizabeth Ann Micks 

Academic Editor

PLOS ONE